# Goal-Conditioned Reinforcement Learning from Sub-Optimal Data on Metric Spaces

## Abstract

We study the problem of learning optimal behavior from sub-optimal datasets for goal-conditioned offline reinforcement learning under sparse rewards, invertible actions and deterministic transitions. To mitigate the effects of *distribution shift*, we propose MetricRL, a method that combines metric learning for value function approximation with weighted imitation learning for policy estimation. MetricRL avoids conservative or behavior-cloning constraints, enabling effective learning even in severely sub-optimal regimes. We introduce distance monotonicity as a key property linking metric representations to optimality and design an objective that explicitly promotes it. Empirically, MetricRL consistently outperforms prior state-of-the-art goal-conditioned RL methods in recovering near-optimal behavior from sub-optimal offline data.

## 1 Introduction

Effective decision-making is an integral part of intelligent behavior. To achieve this, learning-based control methods have proven to be a viable option in complex scenarios (Andrychowicz et al., 2020; Silver et al., 2017; Mnih et al., 2013; Peters & Schaal, 2008). In particular, reinforcement learning (RL) allows learning near-optimal behavior through trial-and-error (Sutton & Barto, 2018). However, the online reinforcement learning framework generally requires slow, expensive (and potentially dangerous) online interactions with the environment.

Offline RL, on the other hand, formalizes the learning of optimal behaviors from a *static* dataset (Levine et al., 2020). This approach offers many advantages over its online counterpart, such as the ability to leverage large-scale datasets to learn complex behavior (Walke et al., 2023; Dasari et al., 2020) without the need of re-collecting data (Shi et al., 2021; Gürtler et al., 2023). Because of the inability to access the environment, in offline RL it is assumed that the dataset already includes suitable

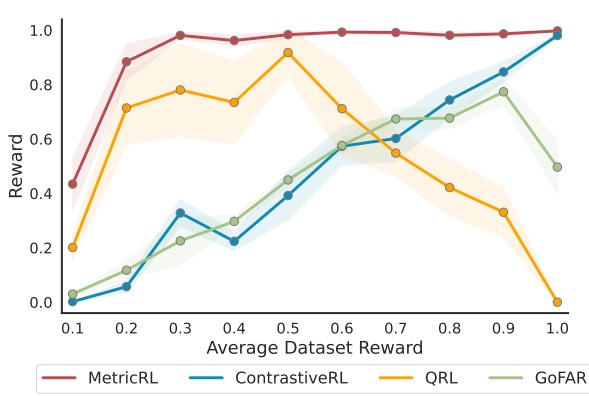

Figure 1: Average reward on `Minigrid DoorKey` (Chevalier-Boisvert et al., 2023) as a function of the expected reward present in the offline dataset. We contribute MetricRL (red line), a novel goal-conditioned offline RL agent able to learn near-optimal behavior from severely sub-optimal datasets.

information to perform the given task. This data, however, may often be collected by sub-optimal agents. In this work, we address the question of how to learn different (and *better*) behaviors from the one observed in the dataset, regardless of its optimality. We focus on the challenge of *learning near-optimal behavior from severely sub-optimal datasets*, such as the ones collected by a random policy. We empirically demonstrate in Section 4, and highlight in Figure 1, how under these conditions current offline RL methods (Eysenbach et al., 2022; Wang et al., 2023; Ma et al., 2022b; Kostrikov et al., 2021) struggle significantly.

We focus on sparse-reward goal-conditioned offline reinforcement learning, which aims at learning optimal behavior to reach multiple goals within the same environment. Motivated by recent work in metric (Park et al., 2024; 2023b) and quasimetric (Eysenbach et al., 2022; Wang et al., 2023) learning for RL, we study an approximation of the optimal value function in severely sub-optimal data conditions without explicitly relying on the Bellman operator, (Bellman, 1954). The core idea is to learn an embedding of the state space such that distances correspond to the minimum number of actions needed to reach one state from the other. We introduce the notion of *distance monotonicity* (DM) as a relaxation of isometric embeddings and show that this property is sufficient to provably recover an optimal actor-critic policy. In particular, we propose to pair this approximation of the value function with a weighted imitation learning actor, a method we refer to as *MetricRL*, to recover an optimal policy *regardless of the dataset quality*. This is made possible by avoiding the out-of-distribution issue caused by the max operator used in dynamic programming solutions, without requiring any additional conservative (e.g., as in CQL (Kumar et al., 2020)) or behavioral (e.g., as in BCQ (Fujimoto et al., 2019), BEAR (Kumar et al., 2019)) regularization.

In this work, we define the learned embedding as a Euclidean metric space. This effectively limits the applicability of the model to reversible processes (Steccanella & Jonsson, 2022). On the other hand, we empirically show that this induced bias can ease the learning process in offline RL for many established environments. We evaluate MetricRL across a wide range of literature-standard goal-conditioned reinforcement learning tasks. We show how MetricRL outperforms prior goal-conditioned offline reinforcement learning methods in learning near-optimal behavior from severely sub-optimal datasets. Additionally, we show how MetricRL easily scales to high-dimensional observations.

In summary, our contributions are the following:

- **MetricRL**: We propose a novel method that exploits symmetries in latent representation spaces for reversible goal-conditioned offline reinforcement learning.

- **Distance Monotonicity**: We define a new property of these latent representation spaces. We show that, under invertible actions, preserving such a property provably leads to policy optimality.

- **Learning from Sub-Optimal Offline Data**: We demonstrate how MetricRL is able to recover optimal behaviors in severely sub-optimal data conditions, outperforming prior state-of-the-art offline RL methods across literature-standard environments.

## 2 Preliminaries and Assumptions

**Goal-Reaching Reinforcement Learning**: We consider standard Markov Decision Processes (MDPs) for goal-reaching tasks, $\mathcal{M} = (S, A, T, r, \gamma)$, where $S$ is the state space, $A$ is the action space, $T : S \times A \to S$ is a deterministic transition function (a common assumption in recent offline RL methods (Ma et al., 2022b; Park et al., 2023a; Wang et al., 2023)), $r : S \times A \to \mathbb{R}$ is a goal-conditioned, sparse reward, i.e., $r(s, a) \neq 0$ iff $T(s, a) = s_g$ (where $s_g$ is the goal-state), and $\gamma \in [0, 1)$ is a discount factor on the future rewards of the agent. We additionally define the goal states to be absorbing and consider the process terminated once these are reached.

The goal of the process is to find an optimal policy $\pi^*(a \mid s, s_g)$ that, given goal state $s_g$, maximizes the cumulative discounted reward of the agent for any possible starting state. To evaluate the optimality of the policy we resort to the goal-conditioned value function $V^\pi(s, s_g)$, defined as the expected discounted cumulative reward starting in a particular state and acting according to a policy $\pi : V^\pi(s, s_g) = \mathbb{E}_\pi \left[ \sum_t \gamma^t r_t | s_0 = s, a = \pi(s, s_g) \right] \quad \forall s \in S$. The value function associated with the optimal policy is referred to as the optimal value function $V^* = V^{\pi^*}$ and, as such, is always greater or equal to any other value functions, i.e., $V^*(s, s_g) \geq V^\pi(s, s_g) \quad \forall s, \pi$.

**Offline Reinforcement Learning**: In the offline RL setting, we assume we have access to a dataset $D$ of interactions with the environment described by the MDP and collected by some unknown policy $\pi_\beta$, i.e., $D_{\pi_\beta} = \{(s, a \sim \pi_\beta(a \mid s), r, s' = T(s, a))\}$. A major issue in Offline RL stems from the way dynamic programming methods estimate the optimal value function of the MDP. Most of the current algorithms rely

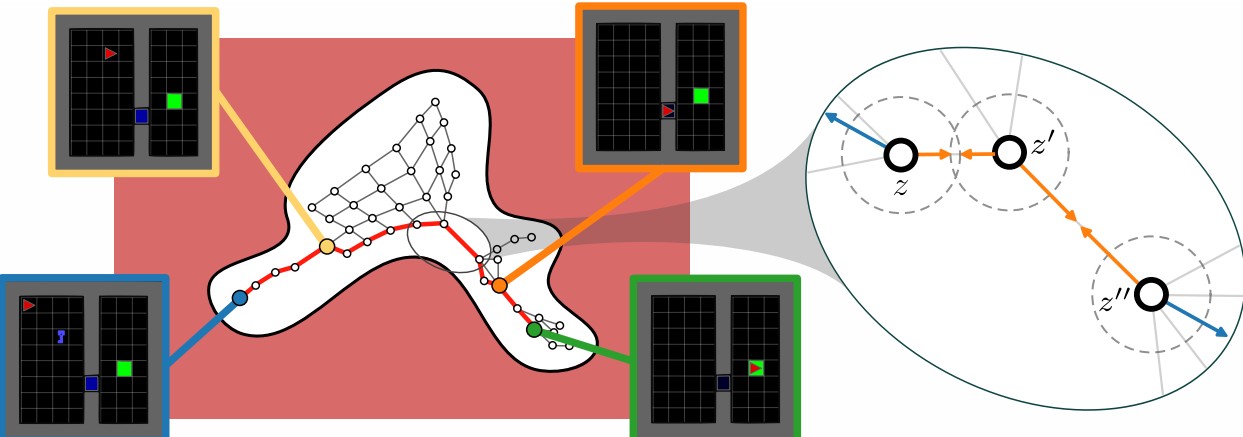

Figure 2: We explore a form of symmetry in representation learning for goal-conditioned offline reinforcement learning: we learn a metric space in which Euclidean distances between the representation of states $(z, z', z'')$ are related to the value function of the agent. We call our approach **MetricRL**. In the `Minigrid Doorkey` environment, moving greedily to adjacent states translates to the optimal policy (red line) to reach the goal (in green). Our objective is to preserve the local structure of adjacent representations (orange arrows, left) while maximizing the separation between non-adjacent ones (blue arrows, left).

on temporal difference techniques to learn the critic function, e.g., DQN (Mnih et al., 2013), CQL (Kumar et al., 2020), BEAR (Kumar et al., 2019), BCQ (Fujimoto et al., 2019). However, the max operator used to estimate the target value has been shown to result in an overestimation of the expected return, Levine et al. (2020). While this is not an issue in Online RL, as the agent has the possibility of exploring overestimated states, it results in catastrophic effects in Offline RL. In this paper, we rely on an alternative technique for learning the critic. As discussed in prior work (Wang et al., 2023; Yang et al., 2020), the dataset $D$ implicitly defines a graph $G = (S, A)$ where the nodes correspond to the states and the edges to the actions of the agent. Here, we assume that this graph only has one connected component, i.e., any two states in the dataset are connected through a path on the graph. Furthermore, we assume that there always exists an *inverse* action, i.e., $\exists a' \in A : T(s', a') = s \quad \forall (s, a, s' = T(s, a))$. Note that $a'$ doesn't necessarily need to correspond to the actual opposite action and it can be any viable action. By endowing the graph $G$ with a metric $d_S$, we can define a corresponding finite metric space $(G, d_S)$. For the tuple to be a valid metric space we need to define the metric $d_S : S \times S \to \mathbb{R}$ to respect the axioms of a metric space (Burago et al., 2001). In particular, we can define the distance between any two states to be the number of edges on the shortest path connecting them (geodesic distance).

## 3 Method

In this work we focus on learning near-optimal goal-conditioned behavior from sub-optimal offline data. In these conditions, two main problems arise. The first is to learn an optimal behavior without depending on the quality of the distribution of the data used. The second is to avoid the out-of-distribution shift commonly faced in offline reinforcement learning (Levine et al., 2020). We propose to address both using a metric learning approach to estimate the optimal value function. To do so, we start by defining *distance monotonicity* (Section 3.1), a novel property on representations needed to recast the problem of optimal value function estimation into metrics. We propose a loss function to learn maps that respect such a property and show how to build an approximation of the value function using distances in this learned representation (Section 3.2). Finally, we define an actor-critic method to learn policies over this approximation and formally prove their optimality (Section 3.3). We call our approach *MetricRL*.

### 3.1 Distance Monotonicity

Consider a continuous map between the state space of an MDP for goal-reaching tasks and a latent representation space: $\phi : S \to Z \subseteq \mathbb{R}^n$. We can equip this latent vector space with a Euclidean norm to obtain a Euclidean metric space $(Z, \|.\|_2)$. Here we consider distances in the original space, $d_S$ as the minimum number of actions an optimal policy needs to reach one state from the other. We say $\phi$ is isometric if relative distances in the original state space $d_S$ and the latent metric space $d_Z$ are preserved, i.e., $d_Z(\phi(s), \phi(s')) = d_S(s, s')$, $\forall s, s' \in S$. In fact, if $\phi$ is isometric then the value function can be defined as simply the norm of the distance between the current state and the goal state, i.e., $V^*(s, s_g) = \gamma^{d_Z(\phi(s), \phi(s'))} r_g$ where $r_g$ represents the reward at the goal and the expectation has been dropped as we assume deterministic transitions. We present an extended discussion of this observation in Appendix A.1.

However, estimating an isometry between these two metric spaces is known to be not always possible (Bourgain, 1985; Matoušek, 2002). To overcome such an issue, we consider a relaxation of isometries between metric spaces. Given two metric spaces $(S, d_S)$ and $(Z, d_Z)$(e.g., $d_S$ the geodesics in the state space and $d_Z$ a simple $\ell_2$-norm) and the corresponding map $\phi$ between them, we can define the following property of $\phi$:

**Definition 3.1.** *We say $\phi$ is* distance monotonic *(DM) if for all $s_1, s_2, s_3 \in S$, the following holds:*

$$d_S(s_1, s_3) < d_S(s_2, s_3) \implies d_Z(\phi(s_1), \phi(s_3)) < d_Z(\phi(s_2), \phi(s_3)).$$

We propose to parameterize the map $\phi_\theta$ and learn it by minimizing the following objective on the dataset $D$:

$$\mathcal{L}_\theta(D) = \mathbb{E}_D \left[ (\|\phi_\theta(s') - \phi_\theta(s)\|_2 - 1)^2 - \lambda \|\phi_\theta(s'') - \phi_\theta(s)\|_2 \right], \tag{1}$$

where $(s, s')$ are any two states connected by an action sampled from $D$ and $s''$ are other states sampled independently from the dataset at random.

Our loss function balances two requirements on the learned representation: the first term forces the representation to preserve the local distances of states in the graph, encouraging connected states to be separated by a vector of norm one in the latent representation $Z$. As we show in Figure 2 (right, orange arrows), the representation of connected states $(z, z')$ lies on a circumference of radius one. The second term of the loss maximizes the distance of non-directly connected states, as highlighted in Figure 2 (right, blue arrows). This term of the loss is unbounded if the graph defined by the dataset is not composed of a single connected component. As we discuss in Section 3.4, in cases where the dataset defines multiple connected components (e.g., with image observations) we can always define a synthetic super-node to connect every termination state.

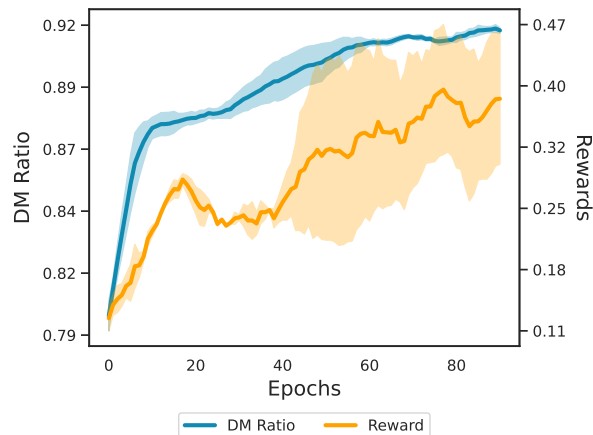

Figure 3: Optimizing Equation 1 increases the ratio of distance monotonic triplets (blue curve) on `Maze2D` (Large). Distance monotonicity is also correlated with an increase in the average return of the agent (orange curve).

The minimization of Equation 1 enforces the learning of a distance monotonic map: in Figure 3 we highlight that the ratio of distance monotonic triplets significantly increases as a function of the training of the map. This can be approximated by discretizing the state space, building an $\epsilon$-graph and the resulting distances in the state space as geodesics on the graph, comparing these distances with distances in the learned representation $Z$. In Appendix A.2 we provide a more formal intuition of this relationship as well as additional details on the evaluation of the distance monotonicity ratio.

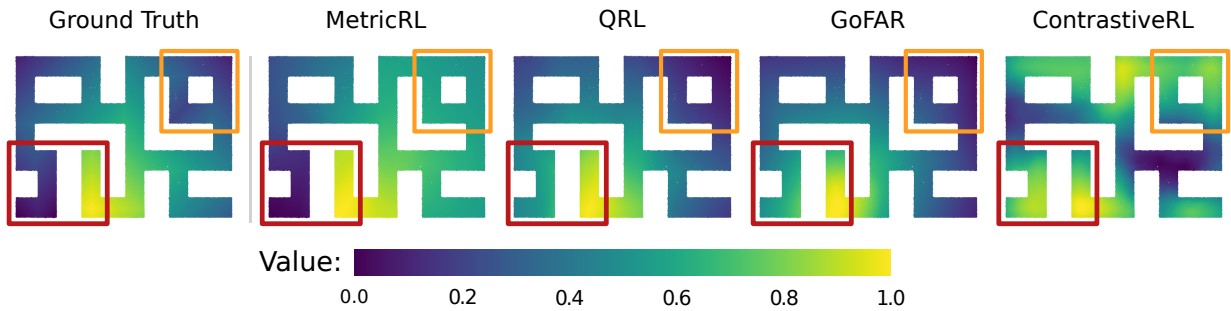

Figure 4: Estimated value function for different methods using a dataset collected from a random policy in Maze2D Large. We highlight (in red) how MetricRL is the only method able to correctly assign low values to states that are close in the Euclidean space but not in terms of *distance to the goal*. Additionally, we highlight (in orange) that our proposed distance monotonicity in complex topologies is not equivalent to isometries, yet we are still able to recover provably optimal policies (as we discuss in Section 3.3). All values are normalized.

## 3.2 Value Function Approximation

Distance monotonic representations (Definition 3.1) allow us to recast the original goal-conditioned RL problem as a distance one: similarly to the case of isometric maps, we can approximate the optimal value function using distances in the learned latent representation:

$$\tilde{V}(s) = \gamma^{d_Z(\phi_\theta(s), \phi_\theta(s_g))} r_g, \tag{2}$$

where $s_g$ is the goal state and $r_g$ is its associated sparse reward (only given at the goal state). This approximation would be identical to the true optimal value function only when the representation is an isometry of the graph.

However, distance monotonicity is enough to retain relative distances between states. Figure 4 shows the estimated value function of different algorithms for a maze-like problem, using offline datasets collected with a random policy. In such problems the value function should retain the topology of the maze and estimate the value of each position in the maze based on the distance to the goal within the maze.

Figure 4 (red) highlights that a distance monotonic representation (MetricRL) is the only value function able to correctly estimate the low value of the bottom left corner of the maze when the goal is on the other branch of the maze. Equations 1 and 2 allow us to abstract the estimate of the value function from the distribution of the policy that collected the dataset, as we highlight in Figure 11 for policies of different values. Note that, for more complex topologies (Figure 4, orange), distance monotonicity is not equivalent to isometries. However, as we show in the next subsection, our approximation still allows us to estimate a provably optimal policy.

## 3.3 MetricRL

Enforcing distance monotonicity in the latent representation allows us to learn an *approximation* of the true value function, not fully recover it. However, when distance monotonicity is preserved, a greedy policy built on this value function is optimal given some assumptions. Define the greedy policy according to the value function $V$ as $\pi_g^V(s) = \arg\max_a V(T(s, a))$, note that in general the argmax results in a set of possible actions. We have the following:

**Theorem 3.2.** *If the MDP is deterministic, sparse, and goal-conditioned, then*

$$\pi_g^{\tilde{V}}(s) \subseteq \pi_g^{V^*}(s), \quad \forall s \in S$$

*holds if $\phi$ is distance monotonic.*

Intuitively, Theorem 3.2 states that every policy greedy on the value function defined in Equation 2 is optimal. A proof can be found in Appendix A.3. When the actions space is too large or continuous, we pair the learned representation with a parametric policy.

To learn such policy we take a three-stage approach, which we call *MetricRL*: (i) we learn a distance monotonic map, using the loss function of Equation 1; (ii) we define the value function using Equation 2; (iii) we estimate a policy using the following loss function:

$$\mathcal{L}_\pi = \mathbb{E}_{s,a \sim D} \left[ -(\tilde{V}(s') - \tilde{V}(s)) \log(\pi(a \mid s)) \right]. \tag{3}$$

This optimization procedure for the policy is akin to the family of weighted maximum likelihood Nair et al. (2020) or weighted imitation learning Wang et al. (2018). A key difference is that the advantage is not exponentiated. Empirically, this didn't have particular effects on the learning of the policy and allowed the removal of the additional hyper-parameter controlling the temperature of the exponential, thus simplifying the training procedure. Using a value function from a distance monotonic representation ensures the term $\tilde{V}(s') - \tilde{V}(s)$ is positive only for actions that bring the agent closer to the goal. This approach for the policy update is particularly suitable for offline RL: actions are sampled from the dataset which guarantees us to consider only in-distribution actions. Moreover, using the proposed distance monotonic representation instead of Temporal Difference methods for the critic solves the classic offline RL issue of out-of-distribution transitions (Levine et al., 2020). We provide a pseudocode of our approach in Appendix A.5.

### 3.4 Practical Implementation

**Stabilizing the loss** In practice, the negative term of the loss 1 is unbounded. In practice, we propose to take the logarithm of this contrastive term. This effectively changes the dynamics of the learning, resulting in a weaker pull from the negative terms. We have found this formulation to stabilize the training, in particular for environments with longer episode horizons:

$$\mathcal{L}_\theta(D) = \mathbb{E}_D \left[ (\|\phi_\theta(s') - \phi_\theta(s)\|_2 - 1)^2 - \lambda \log \|\phi_\theta(s'') - \phi_\theta(s)\|_2 \right]. \tag{4}$$

**Learning with images** When learning with images, often the goal information is present in the image, e.g., the position of the green square in the grid experiment in Figure 5, which can break the connectivity assumption: two images with different goal positions are not connected by any path. This can be a problem in practice as the second term in Equation 1 can grow indefinitely when comparing representations of images with different goals. However, in the case of finite MDPs, we can easily recover it by introducing an additional *super*-state that connects every termination state together. We present the implementation details of such a super-state in Appendix A.4. The introduction of this super-state and the consequent connection of the environments with different goals leads to a modification of the learned representation. Figure 5 shows this learned representation. On the left is the distribution of the states for one single goal, while on the right is the distribution of all the goals connected by the super-state (red star). The solution to this representation is a radial distribution of the states connected in the middle by the super-state. All the states with the same goal (orange dots in the image) compose one ray of the overall distribution.

## 4   Results

We evaluate MetricRL against state-of-the-art baselines in offline reinforcement learning across multiple environments. In all experiments we consider three types of offline datasets: a *low* dataset, often collected using a random policy or an untrained agent; a *medium* dataset, collected using the policy of an online RL agent during training or adding stochasticity to a fully trained agent, and a *high* dataset, collected using the policy of a fully-trained online RL agent.

For baselines, we consider the following: CQL (Kumar et al., 2020), BCQ (Fujimoto et al., 2019), BEAR (Kumar et al., 2019), PLAS (Zhou et al., 2021), IQL (Kostrikov et al., 2021), ContrastiveRL (Eysenbach et al., 2022), QRL (Wang et al., 2023), GoFAR (Ma et al., 2022b), HIQL (Park et al., 2023a). More details in Appendix A.6.

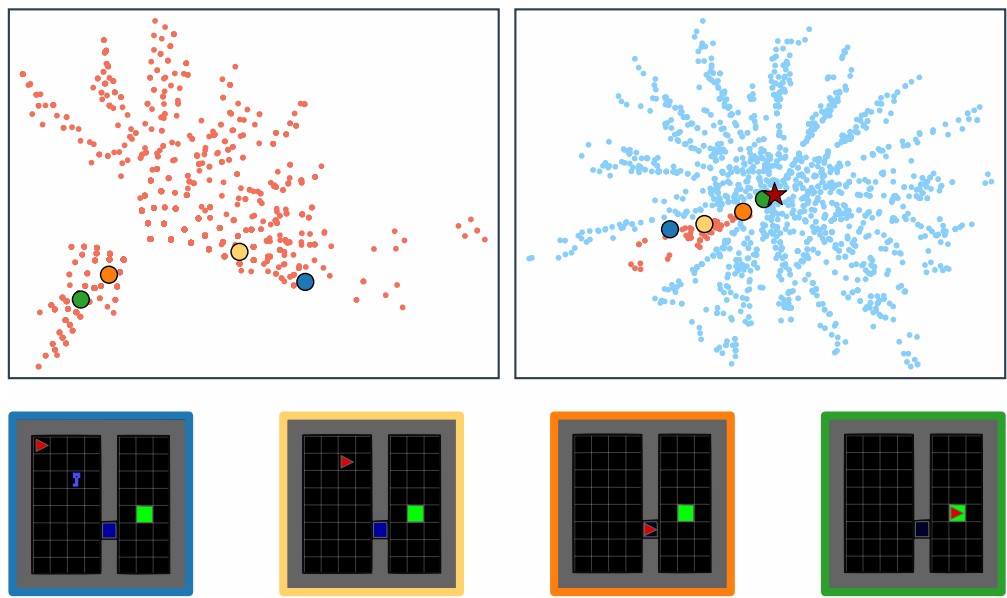

Figure 5: Visualization of the two-dimensional latent space of MetricRL in the `DoorKey` environment when considering state features (left) and image (right) observations. We observe that the addition of a super-state (red star on the right figure) for image observations results in a significant change in the structure of the embedded graph as each set of states with a different (and visible) goal gets separated (orange dots). Nonetheless, in both cases, the optimal policy still follows a geodesic in the graph: from the starting state (blue) the agent needs to pick up the key (yellow) to open the locked door (orange) and move to the goal state (green).

For each model, we perform standard hyperparameter tuning or use the author's suggested hyperparameters (if available). The complete list of training hyperparameters is available in Appendix A.6, CQL, BCQ, BEAR, PLAS, IQL are implemented using Seno & Imai (2022). For the remaining models, we use the author's provided code.

For environments, we consider:

- `Maze2D` (Fu et al., 2020): a navigation task within a two-dimensional maze, with continuous actions and Newtonian physics. We consider three sizes of the maze: *u-maze*, medium and large. For each size, we use a uniform random policy to collect the low dataset, a policy with Ornstein-Uhlenbeck noise (Uhlenbeck & Ornstein, 1930) to collect the medium dataset, and we use the Minari dataset provided in D4RL for the high dataset (Fu et al., 2020);

- `Reach` (Plappert et al., 2018): a manipulation task with a 7-DoF robot with continuous actions to reach a randomly-selected goal position in the workspace. To collect the datasets we employ a PPO agent trained on dense rewards along three different stages of training. For the low dataset, we use the policy of the randomly-initialized agent, for the medium dataset we use a PPO agent achieving half of the optimal expected reward, and for the high dataset we use the policy of the fully-converged agent;

- `Hypermaze`: a novel navigation task on a grid-like $n$-dimensional maze with discrete actions. To collect the offline datasets we employ a DQN agent trained online performing actions using an $\epsilon$-greedy policy: for the low dataset we use purely uniformly random actions, for the medium dataset we sample random actions half of the time and optimal actions the other half, and for the high dataset we use the policy of the fully-trained agent;

- `Minigrid` (Chevalier-Boisvert et al., 2023): a navigation task on a grid-like 2D room with discrete actions. We restrict the action space of the agent to navigation actions and remove rotations. To

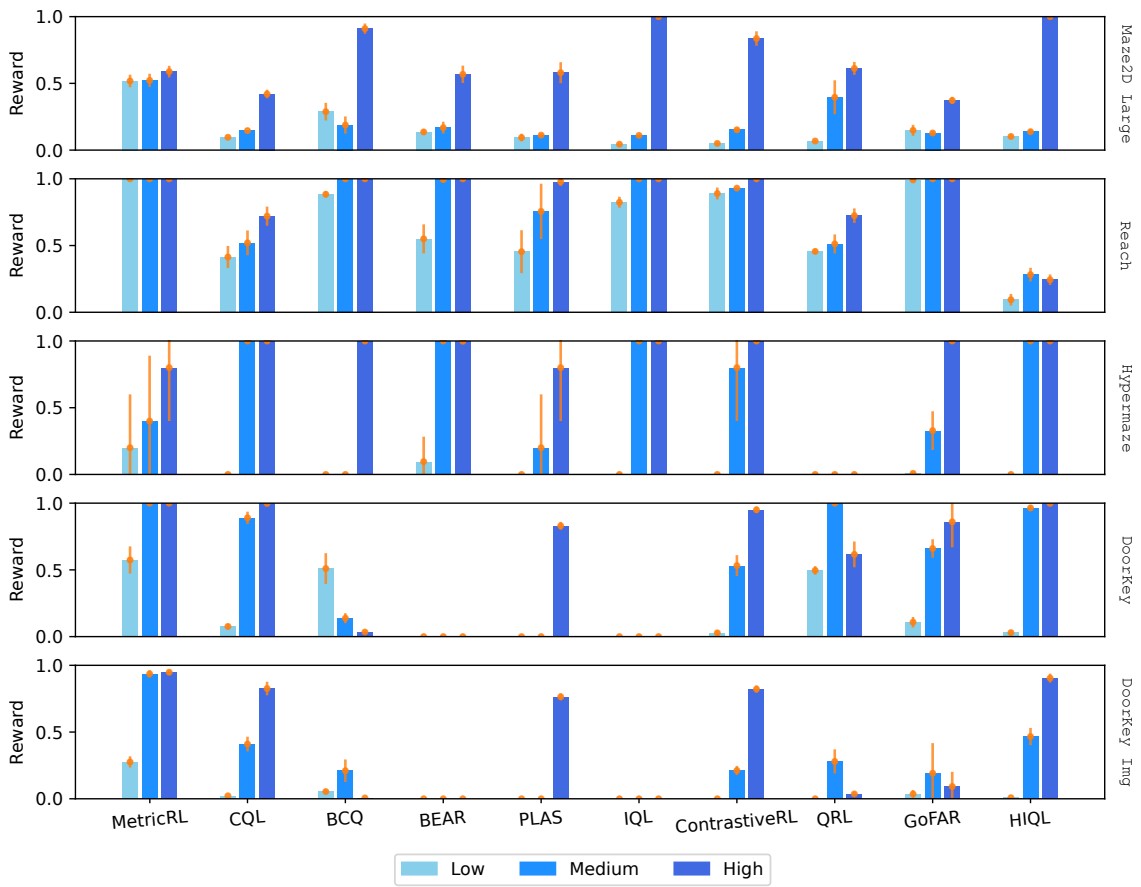

Figure 6: Average reward returns on offline RL tasks with different types of datasets. All results are averaged over 5 randomly-selected seeds. Higher is better. We present extra results in Appendix A.9. MetricRL is the only model able to consistently learn near-optimal behavior from sub-optimal datasets (low and medium), outperforming the baselines, while performing on par with optimal datasets (high).

collect the offline datasets we employ the same strategy as above, that is $\epsilon$-greedy on a fully trained DQN agent. We consider 2 tasks, `Minigrid Empty` consists of an open grid with external walls as the only obstacles. `Minigrid DoorKey` is a three-step task where the agent must first pick-up a key, then open a door, then reach a goal position;

We present our main results in Figure 6[1] For a complete list of hyperparameters employed in the data collection procedure, please refer to Appendix A.6.

### 4.1 Discussion

**MetricRL outperforms other methods in learning from sub-optimal datasets**: The results highlight that MetricRL is the only model able to maintain a consistent level of performance, regardless of the type of dataset used for training. Additionally, MetricRL outperforms the other baseline methods when learning a policy from datasets collected using sub-optimal policies (*low* and *medium* datasets). In particular, for low datasets, MetricRL consistently outperforms all other baselines.

**MetricRL provides more stable training across different data distributions in offline datasets**: We conduct an additional experiment on the `Minigrid DoorKey` environment and consider a finer discretization of the distribution of rewards. We collect multiple datasets using an increasing value of $\epsilon$ for an $\epsilon$-greedy

---

[1]We present in Appendix A.9 additional results. The conclusions remain the same for those additional environments.

optimal DQN agent. As shown in Figure 1, MetricRL (red line) requires datasets with a smaller average reward than the baselines to achieve optimal performance. Additionally, the results show that MetricRL does not suffer from out-of-distribution data when the dataset has a very narrow distribution (almost all or all optimal trajectories).

**MetricRL outperforms QRL in tasks with large action spaces**: MetricRL is able to estimate good policies in both discrete action spaces and continuous ones. In particular, it manages to converge in very high-dimensional action spaces like the `Hypermaze` where there are 81 possible actions with low datasets.

**MetricRL can learn from sub-optimal datasets of images**: To evaluate the performance of MetricRL when provided with high-dimensional observations (images) we reuse the MiniGrid `Empty` and `DoorKey` environments and introduce a *super*-state following the discussion in Section 3.4. For every model, we introduce a CNN architecture that maps the images to a lower-dimensional representation. This highlights that MetricRL maintains its performance, and remains the only model able to learn optimal behavior from sub-optimal datasets. We present additional results in line with this on multiple environments, see Figure 15 in Appendix.

In Appendix A.7 we explore the sample efficiency of MetricRL in scenarios with large state spaces. The results show that our method significantly outperforms temporal-difference (TD) methods (e.g., DQN (Mnih et al., 2015)): to solve larger state space problems, we require a linear increase in the number of training iterations against the exponential increase of TD methods. In Appendix A.8, we show how MetricRL can be used in multi-goal tasks without modifications, considering changing multiple goals and discount factors at execution time.

## 4.2 Metric Space Ablation

We further ablate the impact of the symmetry bias and the use of a Euclidean embedding for value function approximation. Specifically, we compare MetricRL against an equivalent model that employs a quasimetric critic, following Wang et al. (2023). For a fair comparison, we use the same actor optimization as in Equation 3. We also evaluate a symmetrized variant of the quasimetric model by augmenting the dataset with synthetic reverse transitions, i.e., $(s', a, s)$. Results on the `Maze2D`-large

Table 1: Ablation on symmetrizing data in the `Maze2D` Large environment.

| Dataset | QRL | QRL - sym | MetricRL |
|---|---|---|---|
| Low | $0.13 \pm 0.05$ | $0.25 \pm 0.04$ | $0.51 \pm 0.05$ |
| Medium | $0.22 \pm 0.03$ | $0.52 \pm 0.02$ | $0.56 \pm 0.05$ |
| High | $0.28 \pm 0.08$ | $0.56 \pm 0.12$ | $0.60 \pm 0.09$ |

environment (Table 1) show that incorporating symmetry can significantly enhance performance when it reflects a valid inductive bias. Moreover, Euclidean embeddings often prove easier to optimize, particularly in low-quality or sub-optimal data regimes.

## 4.3 Limitations

MetricRL recasts the computation of the value function as a problem of measuring distances in an appropriate learned metric space. To do so, it requires two additional assumptions on the MDPs it is applied to: the existence of inverse actions and the connectivity of the dataset used to learn the value function. As stated in Section 3.4, the connectivity assumption can be solved using "super-states" to join the states into a unique connected component. The existence of inverse actions, on the other hand, defines a trade-off. It limits the applicability of the proposed method to MDPs where the assumption is respected. On the other hand, it greatly simplifies the learning process by imposing a relevant bias on the representation. MetricRL, in fact, requires significantly fewer data points as it can interpolate missing transitions when the inverse transition is present in the data. This allows to estimate effective policies even in severely sub-optimal data conditions.

## 5    Related Work

**Offline RL:** The challenge of learning a policy from a static sub-optimal dataset of transitions and rewards has been extensively studied (Levine et al., 2020). The problem of exploration is not considered as it is assumed that the dataset given contains all the relevant information to estimate an optimal policy. Methods can be roughly divided into four main categories. The first one is constraining the learned policy to not deviate much from the policy that collected the dataset: Kumar et al. (2019) restrict policies to have the same support as the behavior policy rather than the policy itself; Zhou et al. (2021) implicitly constrain the policy by learning it using latent representations of actions with Gaussian prior (VAE), the constraint given by the KL divergence term; Siegel et al. (2020) learns the behavioral policy explicitly. The second category introduces a penalty in the reward function based on the uncertainty of the transitions or the reward function. This penalty has been defined as the uncertainty of the learned $Q$ function, (Kidambi et al., 2020; Yu et al., 2020), or a measure of pessimism (regularization of the highest $Q$) (Kumar et al., 2020). A third family of methods instead uses model-based RL and explicitly computes a model of the environment that can be used in different forms to regularize the learned policy (Matsushima et al., 2020; Yu et al., 2021; Rigter et al., 2022; Fujimoto et al., 2019). The last category includes *in-sample* algorithms which restrict the learning of the policy only on data within the provided dataset and reweighted by an estimate of the advantage function. This family of methods has been referred to as weighted supervised learning (Wang et al., 2018) or maximum likelihood (Nair et al., 2020). It has been extended with different forms of regularization including expectile regression for the critic (Kostrikov et al., 2021), penalizing out-of-distribution actions in the critic (Chebotar et al., 2021), trust regions (Mao et al., 2023), goal relabeling (Yang et al., 2022) and Generative Adversarial Networks (Wang et al., 2024). The policy estimate of our proposed method falls in this last category, as can be seen in Equation 3. The use of Temporal Difference learning for the critic, however, subjects these methods to the problem of distribution shift in Offline RL. In this work, we propose a method that can estimate a high-return policy independently from the distribution of the data used even when the quality of the data decreases substantially.

**Contrastive Learning:** Representation learning techniques have been used to aid the RL problem. Several works have used contrastive learning models to speed up or improve the generalization of a classic RL algorithm, (Laskin et al., 2020; Oord et al., 2018; Anand et al., 2019; Stooke et al., 2021). Other applications include reward function estimation from demonstrations (Ma et al., 2022a; Sermanet et al., 2018) or generating intermediate goals for curriculum learning (Venkattaramanujam et al., 2019). In Eysenbach et al. (2022); Hatch et al. (2023) contrastive learning is used to estimate the discounted state occupancy measure which is equivalent to the Q function in some particular cases. This method however works only when the reward function can be expressed as a goal reaching density and doesn't estimate the optimal Q value but rather the Q value of a current policy, thus still requiring the concurrent learning of a policy in a classic RL fashion. Zhu et al. (2022) proposes the use of contrastive learning to build a representation of states where distances are correlated with reachability in terms of actions. After the representation is learned, they propose to explicitly build the graph of the offline dataset and apply value iteration to get an estimate of the value function. The policy can be obtained by applying Dijkstra on the learned graph. More similar to this work Wang et al. (2023); Yang et al. (2020) estimate the optimal value function rather than the policy one. In Yang et al. (2020) the authors use contrastive learning to find a representation where connections between states in action terms can be estimated in terms of Euclidean distances. The value function can then be recovered as the sum of the shortest path distances between the goal and the current state using classical depth-first search algorithms. Wang et al. (2023) learns a map between pairs of states and a quasimetric representing the estimated optimal value function of a goal-conditioned MDP. This is done by setting the distance between two consecutive states to be equal to the reward between them and the distance between random pairs of states to be maximized. The optimal value function can then be defined as the distance between each state and the goal state. While being more general, quasimetrics cannot capture the appropriate bias induced by the inverse actions assumption. We show empirically that our proposed method is more effective in estimating a policy when data is severely sub-optimal. In this paper, we describe a property of representations needed to ensure the optimality of the critic function and propose a simple method to recover such a representation for a particular class of MDPs. Moreover, differently from the works

above, we study the effectiveness of this methodology in handling offline datasets collected by policies that are not necessarily optimal.

**Other metrics:** Other measures of state similarity from a control perspective have been explored before. Bisimulation metric defines a measure of similarity between states in terms of future transitions and rewards, (Ferns & Precup, 2014; Castro, 2020). These methods are theoretically grounded but particularly difficult to make them work in practice. This is especially true in the case of continuous spaces. Older work has explored different forms of value function approximation. By parametrically approximating the map between each state and its value, (Ormoneit & Sen, 2002) approximates a notion of similarity (in a value sense) between states with an appropriate kernel and rewrite the Bellman operator as a function of this kernel. This still needs to solve the optimization problem with value iteration techniques. Proto-value functions, (Parr et al., 2008), instead express the transition and reward functions as linear matrices, the value function problem has exact solutions and can be estimated with an appropriate kernel method at the cost of expressivity.

## 6 Conclusions

In this paper, we proposed MetricRL, a novel approximation method for the optimal value function of sparse, deterministic, goal-conditioned MDPs. MetricRL relies on learning a *distance monotonic* representation of the state space, allowing it to define a value function that is correlated with the distance of each state to the goal. We have proved that, when the representation is indeed *distance monotonic*, a greedy policy on this approximated value function is optimal for the class of MDPs stated above. Experimentally, we have shown that MetricRL outperforms prior offline RL methods in learning near-optimal behavior from severely sub-optimal datasets. For future work, we plan on generalizing the notion of *distance monotonicity* to quasimetrics (Durugkar et al., 2021; Durugkar, 2023), extending our method to stochastic MDPs and adapting it to online reinforcement learning problems.

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

## A Appendix

### A.1 Sparse Goal-Conditioned Value Functions in Isometric Spaces

Recall the definition of the optimal value function described above as the maximum over the possible policies of the expectation of the discounted cumulative reward: $V^*(s, s_g) = \max_\pi \mathbb{E}_\pi \left[ \sum_t \gamma^t r_t | s_0 = s, a = \pi(s, s_g) \right]$.

In the particular case of a deterministic MDP with a sparse goal-conditioned reward function, the value function simplifies as there is only one state for which the reward is non-zero (the goal state). Moreover, in the optimal value function, the estimated discounted cumulative reward becomes equivalent to the reward at the goal discounted by $\gamma^{d_G(s, s_g)}$, where $d_G(s, s_g)$ is the geodesic. In this setting, the optimal value function estimation can be reduced to a shortest path estimation problem.

Assuming $\phi$ is an isometry, $d_G(s, s_g) = d_Z(\phi(s), \phi(s'))$ resulting in: $V^*(s, s_g) = \gamma^{d_Z(\phi(s), \phi(s'))} r_g$.

### A.2 Distance Monotonicity Measure

We provide a more formal intuition on the increase in distance monotonicity of a representation that minimizes Equation 1. As such, consider the following modification of the objective:

$$\min_\theta \mathbb{E}_D \left[ - \|\phi_\theta(s'') - \phi_\theta(s)\|_2 \right] \tag{5}$$

$$\text{subject to: } \|\phi_\theta(s') - \phi_\theta(s)\|_2 = 1. \tag{6}$$

This can be seen as the second term of the loss in Equation 1 with the first term as an explicit constraint. As defined before, the distance between any two points, $s_i, s_j$, is the length of the geodesic on the graph defined as the offline dataset. This is equivalent to the sum of the intermediate steps within the geodesic path. Using the constraint in Equation A.2 we can use triangular inequality to bound the distance between points in the learned representation:

$$d_Z(z_i, z_j) = \|\phi_\theta(s_j) - \phi_\theta(s_i)\|_2 \in [0, d_S(s_i, s_j)]. \tag{7}$$

The representation's $\phi_\theta$ distance monotonicity can be measured as the ratio of triplets that respect the definition 3.1. That is, if $d_S(s_1, s_3) \leq d_S(s_2, s_3)$ then $d_Z(z_1, z_3) \leq d_Z(z_2, z_3)$, where $z = \phi(s)$. The distance monotonicity of $\phi$ increases for each triplet for which this becomes true. Graphically this can be seen in Figure 7. As the distance $d_Z(z_1, z_3)$ is bounded, the distance monotonicity of $\phi$ increases when $d_Z(z_2, z_3) \in [d_Z(z_1, z_3), d_S(s_2, s_3)]$ (or equivalently $z_2$ goes outside the blue circumference in the figure). As such, distance monotonicity increases by stretching the distance between any two states. This is equivalent to the objective described in Equation A.2.

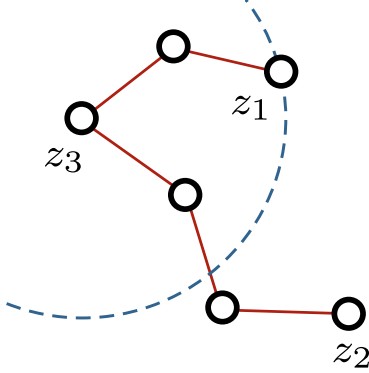

Figure 7: The optimization in A.2 increases the distance between the representations. As long as $z_2$ is outside of the circumference, the triplet $s_{\{1,2,3\}}$ is distance monotonic.

#### A.2.1 Experiment on Distance Monotonicity Measure

We tested quantitatively the effects of minimizing the loss defined in Equation 1 and the increase in distance monotonicity of the learned representation. To do so we used the environment defined in `Maze2D` large with the high dataset. Figure 3 (blue curve) shows the increase in distance monotonicity of a representation throughout the learning process. We computed the measure as follows:

- Before starting the training we compute a discretization of the positions of the maze and an adjacency matrix based on whether two positions are connected or not. This effectively defines a graph of the states of the maze.

- During every epoch of training we sample 1000 triplets of these nodes $(s_1, s_2, s_3)$ at random and compute the distances on the graph between the two pairs using breath-first search, i.e., $d_G(s_1, s_3)$ and $d_G(s_2, s_3)$.

- Concurrently we map these three points in $Z$ using the representation that's being learned and compute the distances using a Euclidean norm, i.e., $d_Z(\phi(s_1), \phi(s_3))$ and $d_Z(\phi(s_2), \phi(s_3))$.

- We then compute the distance monotonicity (DM) ratio as the number of triplets among these 1000 for which if $d_G(s_1, s_3) < d_G(s_2, s_3)$ then $d_Z(\phi(s_1), \phi(s_3)) < d_Z(\phi(s_2), \phi(s_3))$ or if $d_G(s_2, s_3) < d_G(s_1, s_3)$ then $d_Z(\phi(s_2), \phi(s_3)) < d_Z(\phi(s_1), \phi(s_3))$.

- The plot is also paired with the average reward a MetricRL agent achieves during training (orange curve).

More results on this are provided in Figure 16.

### A.3 Proof of Theorem 3.2

*Proof.* Here we prove that, assuming a deterministic, sparse, goal-conditioned MDP, every optimal solution defined as the greedy policy based on the value function defined as in Equation 2 is contained in the greedy policy based on the optimal value function.

We start by rewriting the statement of the Theorem in the argmax form:

$$\pi_g^{\tilde{V}}(s) = \arg\max_a \tilde{V}(s' = T(s, a)), \tag{8}$$

$$\pi_g^{V^*}(s) = \arg\max_a V^*(s' = T(s, a)). \tag{9}$$

Given that the MDP is deterministic, the transition function $T(s, a)$ injectively maps $(s, a)$ to a unique next state $s'$.

From the definition of $\tilde{V}$ and the Bellman equation, we have:

$$\tilde{V}(s') = \gamma^{d_Z(\phi(s'), \phi(s_g))} r_g, \tag{10}$$

$$V^*(s') = \max_{a'} \left[ r(s', a') + \gamma V^*(s'') \right], \tag{11}$$

where $s'' = T(s', a')$. In a sparse, goal-conditioned MDP, the reward $r(s', a')$ is zero unless $s' = s_g$, $V^*(s')$ is equal to the discount factor raised to the power of the minimal number of actions needed to reach the goal and thus simplifies to:

$$V^*(s') = \gamma^{d_S(s', s_g)} r_g. \tag{12}$$

Substituting the value functions into the greedy policy definitions, we get:

$$\arg\max_a \gamma^{d_Z(\phi(s'), \phi(s_g))} r_g \subseteq \arg\max_a \gamma^{d_S(s', s_g)} r_g. \tag{13}$$

Since $\gamma < 1$ and $r_g$ is constant, this reduces to:

$$\arg\min_a d_Z(\phi(s'), \phi(s_g)) \subseteq \arg\min_a d_S(s', s_g). \tag{14}$$

Assume, by contradiction, that there exists a state $\hat{s} = T(s, \hat{a})$ such that the action $\hat{a}$ is in $\pi_g^{\tilde{V}}(s)$ but not in $\pi_g^{V^*}(s)$. This implies:

$$d_Z(\phi(\hat{s}), \phi(s_g)) \leq d_Z(\phi(s'), \phi(s_g)), \quad \forall s' \in T(s, \cdot), \tag{15}$$

$$d_S(\hat{s}, s_g) > d_S(\check{s}, s_g), \quad \text{for some } \check{s} \in T(s, \cdot). \tag{16}$$

By the distance monotonicity of $\phi$, $d_S(\hat{s}, s_g) > d_S(\check{s}, s_g)$ implies $d_Z(\phi(\hat{s}), \phi(s_g)) > d_Z(\phi(\check{s}), \phi(s_g))$, which contradicts the assumption that $\hat{a} \in \pi_g^{\tilde{V}}(s)$.

Thus, the actions minimizing $d_Z(\phi(s'), \phi(s_g))$ are contained in the actions minimizing $d_S(s', s_g)$, implying:

$$\pi_g^{\tilde{V}}(s) \subseteq \pi_g^{V^*}(s), \quad \forall s \in S. \tag{17}$$

This completes the proof.

$\square$

## A.4 Incorporating Super-States in the Dataset

Here we provide a short description on how to add super-states in the dataset to connect it. The steps can be summarized as follows:

- **Define super-states:** These can be defined synthetically by creating a state that is not present in the dataset. In the experiments, we always defined it as a vector of all zeros of the same dimensionality of the state space.

- **Add transitions:** The offline dataset can then be augmented with synthetic transitions. For each trajectory in the dataset, we can append a new transition from the terminating state to the meta state. The action connecting these states is not relevant as it will not be used in the representation. In the experiments, we set the action value to a random (but valid) value.

## A.5 Pseudocode

We provide the pseudocode of our approach below.

---
**Algorithm 1** MetricRL.

---
**Require:** Initialize $\theta, \psi$
**Require:** Offline dataset $B$, hyperparameters $\lambda$, $\eta$
1: **repeat**
2:      Sample batch $(s, a, s', s_g)_{\times B} \sim D$
3:      $s_r = \texttt{shuffle}(s)$                                                     $\triangleright$ shuffle states in the batch
4:      $\mathcal{L}_\theta = (\|\phi_\theta(s) - \phi_\theta(s')\|_2 - 1)^2 - \lambda \log(\|\phi_\theta(s) - \phi_\theta(s_r)\|_2)$
5:      Update $\theta \leftarrow \theta - \eta \nabla_\theta \mathcal{L}_\theta$
6:      $V = \|\phi_\theta(s) - \phi_\theta(s_g)\|_2$
7:      $V' = \|\phi_\theta(s') - \phi_\theta(s_g)\|_2$
8:      $\mathcal{L}_\pi = -(V' - V) \log(\pi_\psi(a|s))$
9:      Update $\psi \leftarrow \psi - \eta \nabla_\psi \mathcal{L}_\pi$
10: **until** convergence

---

## A.6 Experimental Details

For each experiment, we provide the results for 5 runs with different seeds. Each model is trained for 100 epochs consisting of 500 batches of 256 data points each. Every model is trained using the Adam optimizer with a learning rate of $10^{-3}$. All experiments have been conducted using an NVIDIA RTX 3080 GPU accelerator.

Both the policy and the value function are parameterized using a simple Multi-Layer Perception architecture consisting of 3 layers with 64 neurons each and a ReLU activation function. In the case of *MetricRL*, the policy outputs the mean of a Gaussian distribution with fixed variance when the actions are continuous and the logits of a Categorical distribution when the actions are discrete. When the observations are images we use a CNN architecture consisting of 4 layers with 64 filters each of size 3 by 3 to preprocess the images.

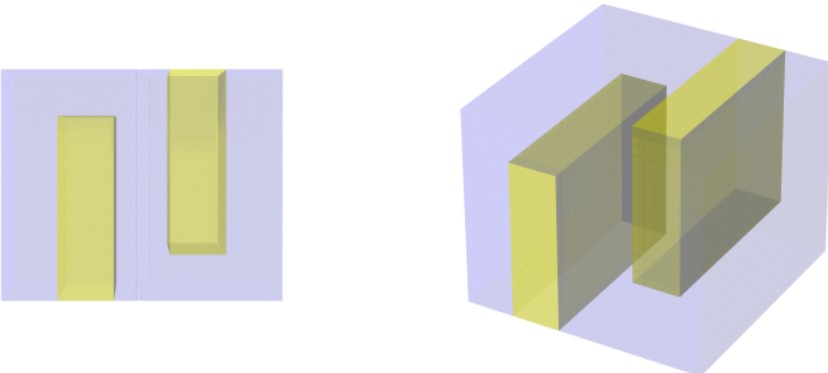

Figure 8: Wall positions (yellow) in the HyperMazes in 2D and 3D, in blue are free cells.

### A.6.1 Models Description and Hyperparameters

**MetricRL**: The representation of the metric space has always dimension 128. The regularization term $\lambda$ in Equation 1 is 1 and the variance of the policy when actions are continuous is 1.

**CQL** (Kumar et al., 2020), which introduces a conservative term in the estimation of the Q value. This term penalizes the highest values of the estimated Q function; hyperparameters: $\gamma = 0.95$, we use a conservative weight of 5.0.

**BCQ** (Fujimoto et al., 2019), which perturbs the policy learned with a VAE with a DDPG term; hyperparameters: $\gamma = 0.95$, action flexibility: 0.5, we sample 10 actions per step and use 2 critics.

**BEAR** (Kumar et al., 2019), that constrains the learned policy to the behavioral one estimated with BC; hyperparameters: $\gamma = 0.95$, we use an adaptive $\alpha$ with an initial value of 0.001 and a threshold of 0.05, we use 2 critics modules and sample 10 actions per step.

**PLAS** (Zhou et al., 2021), which trains the policy within the latent space of a conditional VAE trained on the Offline dataset; hyperparameters: $\gamma = 0.95$, we use 2 critics modules.

**IQL** (Kostrikov et al., 2021), that avoids sampling out-of-distribution actions using a SARSA like critic update with quantile regression; hyperparameters: $\gamma = 0.95$ , we use 2 critics modules and an expectile value of 0.9.

**ContrastiveRL** (Eysenbach et al., 2022), which approximates the value function of the policy that collected the dataset using contrastive learning. To adapt it to offline RL, the objective is coupled with a behavioral cloning term. hyperparameters: $\gamma = 0.95$, an offline regularization of 0.05 and a fixed variance for the policy of 1.

**QRL** (Wang et al., 2023), that estimates a quasimetric to approximate the value function using a contrastive learning formulation. The model is paired with a learned policy regularized with a behavioral cloning term to avoid out-of-distribution state-actions pairs in offline RL conditions. hyperparameters: $\epsilon = 0.25$, initial $\lambda = 0.01$, offset softplus 500 and $\beta = 0.01$ and an offline regularization of 0.05.

**GoFAR** (Ma et al., 2022b), that estimates an offline goal-conditioned RL policy by recasting it as a state occupancy matching problem. hyperparameters: $\gamma = 0.98$ and a discriminator gradient penalty of 0.01. For the f-divergence we use the $\chi^2$-divergence.

**HIQL** (Park et al., 2023a), which reformulates IQL to an action-free hierarchical model. hyperparameters: $\gamma = 0.99$ $\beta = 1.0$, we use 2 critics modules and an expectile value of 0.7.

### A.6.2 Evaluation Scenarios

Maze2D: The goal of the environment is to navigate an agent (actuated ball) to a target position inside a maze of varying complexity. The state space is 4-dimensional consisting of position and velocity in the plane, the action space is the torque applied to the ball in the two directions. The reward function is defined to be 1 when the agent reaches a position within 5 cm from the goal state and zero otherwise. Transitions take into consideration the momentum of the ball, frictions in the environment and collisions with the walls. For the low dataset we collect a dataset of 1000 trajectories of the agent performing uniform random actions, on average around 5% of successful trajectories. For the medium dataset we collect actions according to an Ornstein-Uhlenbeck process with parameters: $\theta = 0.1$ and $\sigma = 0.2$, on average 10% of successful trajectories. For the high dataset we rely on the Minari dataset provided by Younis et al. (2024), 99% successful trajectories.

Reach: The task consists of moving the end-effector of a simulated 7-DoF arm to a desired position in 3D. The state space consists of positions and velocities in 3D of the end effector and gripper of the robot and the goal refers to the desired position of the end-effector and zero velocity. Actions are translations of the end-effector. We first train a PPO agent online to solve the task and collect datasets at different stages of the training to define different qualities. Specifically, low has 20% successful trajectories, medium 60% and high 90%.

Hypermaze: Defines a generalization of the classic Grid Maze navigation task. The environment consists of a hypercube of $n$ dimensions of $m$ cells per dimension where every cell can either be empty or wall. The agent occupies one cell at a time if it is empty and can translate to adjacent cells if they are not walls. The positions of the wall are initialized in an S-like shape similar to Figure 8 when the hypermaze is defined in either 2 or 3 dimensions. The goal of the environment is to reach a goal placed on the other side of the maze.

For the results in Figure 6 we fix the maze to be 4 dimensional with 20 cells per dimension. We collect the datasets by first training a DQN agent online to solve the task and then collect 3 datasets using an $\epsilon$-greedy policy with $\epsilon$ respectively of values 0.9 (success rate 2%), 0.5 (success rate 60%) and 0.1 (success rate 99%).

For the sample complexity analysis our method is coupled with a learned transition function to recover the Q estimate. We vary the dimensions (from 2 to 5) and the number of cells (from 10 to 50). Here the state space is defined as the position of the agent in the maze discretized into cells plus whether there are obstacles or not in the adjacent cells. To train the agents, random states and actions are sampled from the environment. A reward of 1 is given only if the agent steps into the goal state at the end of the maze.

Minigrid: We experiment with two variations of the minigrid environment. The first is the Empty environment where an agent translates freely within a grid-like environment. The state is described by the 2D position of the agent and the actions are the 4 possible translation directions. The reward is 1 once a randomly selected cell is reached and 0 otherwise. The DoorKey environment introduces bottlenecks in the MDP. A wall is introduced in the center of the grid separating it into 2 rooms with a closed door in the middle. In the first room, a key is placed in a random position. The goal is to pick up the key, open the door and reach a goal cell in the other room. The state space is defined as the position in the grid of the agent plus the position of the key plus a binary value describing whether the door is open or not. Actions are translations in the grid plus a pick-up action that has an effect only when the agent is adjacent to the key plus an open door action that has an effect only when the agent has the key and is adjacent to the door. As before, The datasets are collected by training a DQN agent online to convergence and collecting the datasets using $\epsilon$-greedy strategy with the same three different values for $\epsilon$ of before. The success rate is respectively of 70%, 100% and 100% for the Empty environment and 10%, 100% and 100% for the DoorKey environment. Notice how the main difference between the medium and high datasets here is not in the success rate but rather in the quality of the trajectories. For the high-dimensional observations case we use images rendered by the environment as the states. These are 80 by 80 pixels with 3 color channels.

### A.7 Sample-Efficiency

A main advantage of MetricRL stems from the nature of the loss function. Temporal difference methods (e.g., DQN) are known for their inefficiency when the time horizon grows considerably (Moore & Atkeson,

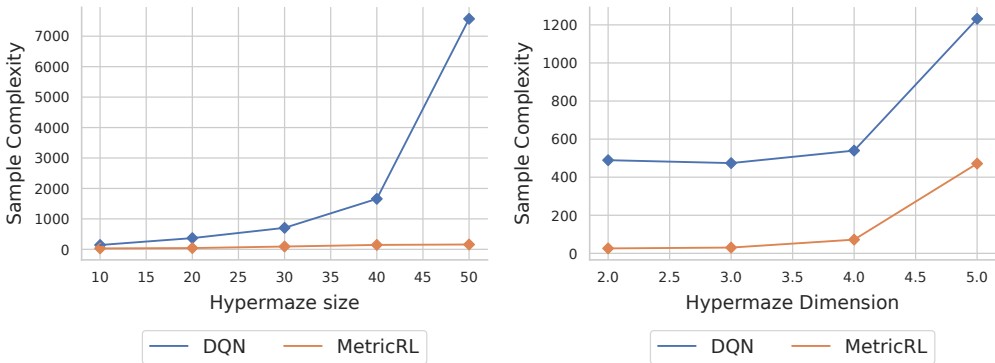

Figure 9: Number of updates required to reliably solve the `Hypermaze` environment with varying number of cells (left plot) and dimensions (right plot).

1993). On the other hand, MetricRL mitigates this issue by using neural networks to learn a representation of a metric space.

To validate our hypothesis, we compare MetricRL against DQN (Mnih et al., 2015) in the `Hypermaze` environment, considering a variable number of dimensions and cells. This allows us to control for both the dimensionality of the action space and the size of the state space of the underlying MDP.

In Figure 9 we present the number of iterations required for the two methods to solve the maze at least 25 times consecutively during training as a function of the size of the maze (state space). The results show that for MetricRL the number of iterations required to learn to perform the maze grows linearly with the size of the maze. However, for DQN the number of iterations rises exponentially.

## A.8 Multi-goal Tasks

Another advantage of the proposed representation used in Equation 2 is that it does not depend explicitly on the goal. As long as goals are valid states within the training distribution, they can be arbitrarily used to estimate the value function. The explicit use of the discount factor allows for reshaping the value function to increase or decrease long-term reward delay. Moreover, with a slight modification, we can easily consider multiple competing goals. In this setting, the agent has to consider both the possible reward it can get in each goal state as well as its relative distance to each goal. As such, the discount factor of the MDP influences the optimal policy of the agent. For example when the agent has to navigate in a maze toward a door but there are multiple doors. In this case, the agent has to learn both how to navigate the maze as well as make the decision of which door to choose based on their reward and the length of the path.

With multiple rewards, the value function approximation can be reformulated by considering the maximum over the value function of all the possible goals $(s_i, r_i)$:

$$\tilde{V}(s) = \max_i \{\gamma^{d_Z(\phi(s), \phi(s_i))} r_i\}. \tag{18}$$

## A.9 Additional Results

In this section, we provide additional results on the experiments described.

**Value Function Estimation in `Maze2D` Large**: Following the discussion of Section 3.2, we explore value function estimation in `Maze2D` Large for different types of datasets. The results of Figure 11 highlight that MetricRL is the only method able to correctly estimate the low value of the bottom left corner of the maze when the goal is on the other branch of the maze, regardless of the quality of the policy used to collect the offline dataset.

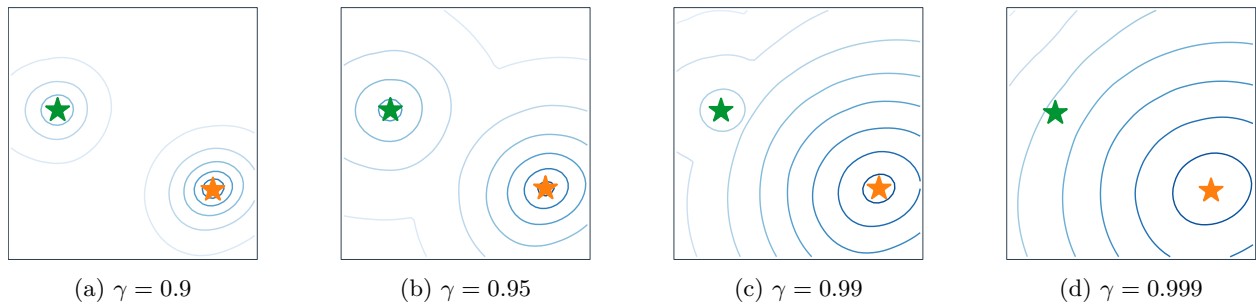

(a) $\gamma = 0.9$       (b) $\gamma = 0.95$       (c) $\gamma = 0.99$       (d) $\gamma = 0.999$

Figure 10: Visualization of the gradient of the value function learned by MetricRL in an environment containing two fixed goals with different rewards $(r_1, r_2)$, as a function of the discount factor: the green star has $r_1 = 0.7$ and the orange star has $r_2 = 1.0$.

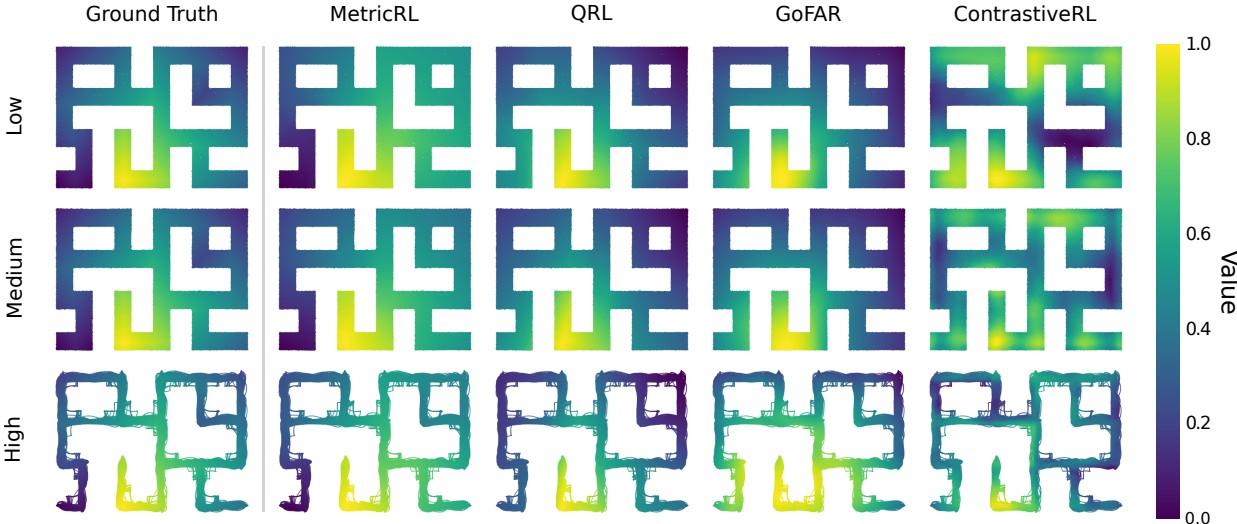

Figure 11: Estimated value function for different methods using a dataset collected from different policies in `Maze2D` Large. All values are normalized.

**Maze2D**: We additionally provide the results for the u-maze and the medium maze described in Fu et al. (2020), (Figures 12 and 13). Results confirm the findings described in the paper. MetricRL consistently outperforms the baselines in the case of low and medium datasets.

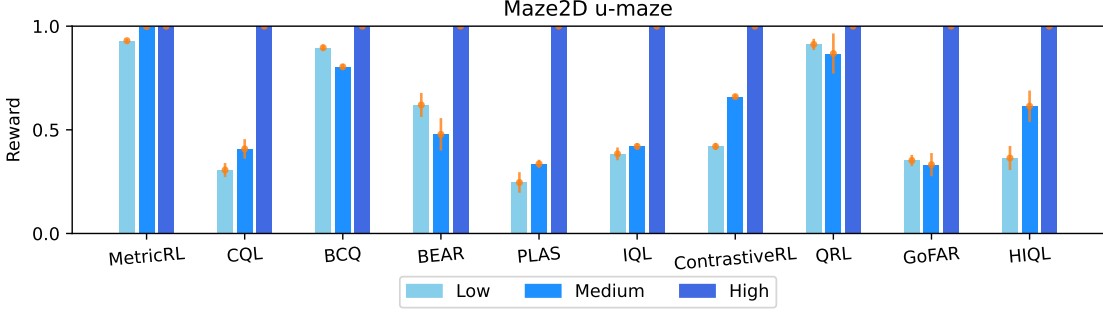

Figure 12: Average reward returns on `Maze2D` u-maze with different types of datasets. All results are averaged over 5 randomly selected seeds. Higher is better.

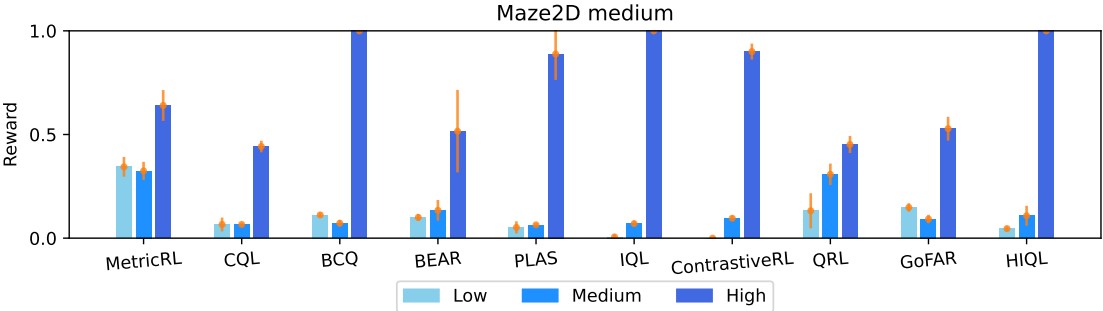

Figure 13: Average reward returns on `Maze2D` medium maze with different types of datasets. All results are averaged over 5 randomly selected seeds. Higher is better.

**Minigrid**: We provide results of the Empty environment with states and images as input, (Figures 14 and 15).

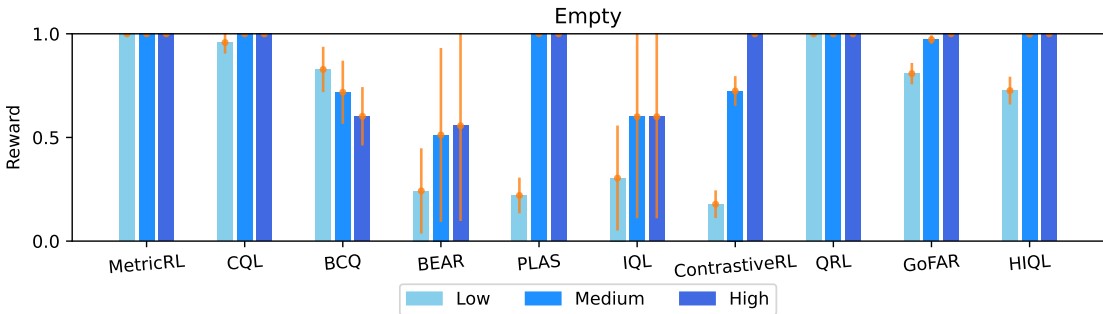

Figure 14: Average reward returns on `Minigrid Empty` with different types of datasets. All results are averaged over 5 randomly selected seeds. Higher is better.

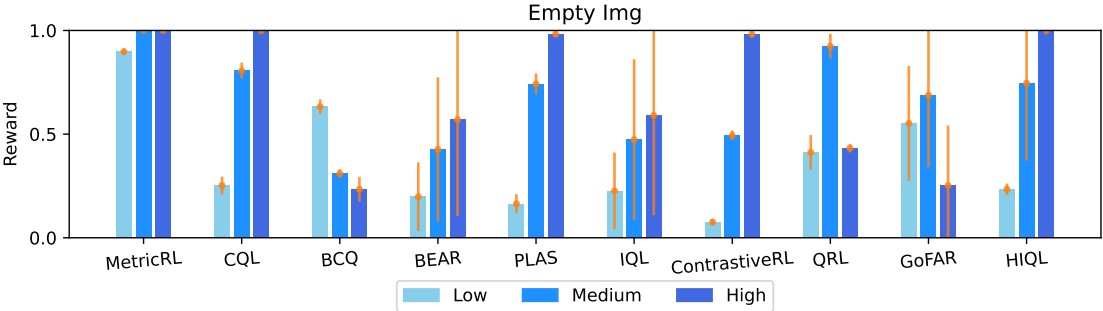

Figure 15: Average reward returns on `Minigrid Empty` with images with different types of datasets. All results are averaged over 5 randomly selected seeds. Higher is better.

Below (Figure 16) are additional results on the increase in distance monotonicity measure paired with an increase in the reward for different mazes and datasets of `Maze2D`.

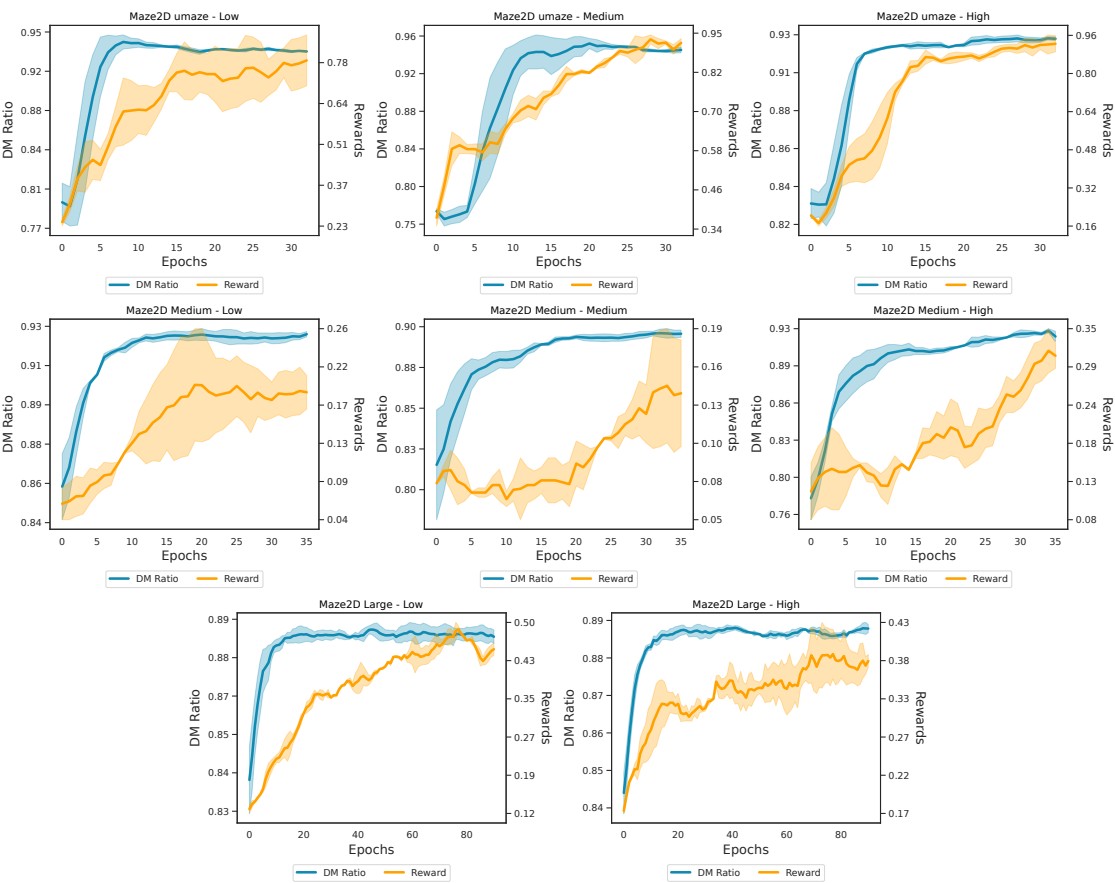

Figure 16: Distance monotonicity ratio compared with average reward on `Maze2D` environments with different mazes and datasets.

