# OpenReview forum: "Goal-Conditioned Reinforcement Learning from Sub-Optimal Data on Metric Spaces"
_TMLR — Rejected by TMLR_

### Review · Reviewer_XvG4 · 2025-12-09

**Summary Of Contributions:**

This paper tackles the problem of goal-conditioned offline reinforcement learning, aiming for a new approach that could better (1) learn from sub-optimal data and (2) reduce or avoid out-of-distribution transitions from the learned policy. The new method, MetricRL, combines weighted imitation learning for policy learning, with a metric learning approach for value functions estimation.

Strengths:

- The proposed methods are clear and the paper is overall well written.
- Assumptions, although quite restrictive, are clearly mentioned and motivated.
- The notion of a distance monotonic map as a relaxation of the strong isometry property is interesting.
- The overall method remains relatively simple and seems promising for goal-conditioned offline imitation learning.

Weaknesses:

- The paper provides theory to motivate MetricRL, but lacks formal arguments to justify the intuition and the claims made.
- Other methods for GCIL based on the Bellman optimality principle exist. It would be interesting to understand how the proposed approach compares to e.g. [1], and how the proposed approaches for avoiding value overestimation compare.

[1] Seohong Park, Aditya Oberai, Pranav Atreya, Sergey Levine. "Transitive RL: Value Learning via Divide and Conquer”. https://arxiv.org/abs/2510.22512

**Audience:**

Yes

**Audience Explanation:**

The paper presents promising new ideas for goal-conditioned offline reinforcement learning, which is of particular interest to the community, including theorists and practitioners.

**Broader Impact Concerns:**

There are no ethical concerns associated with this work.

**Claims And Evidence:**

No

**Claims Explanation:**

The paper presents experiments that seem to support the claims made. However, while theoretical arguments are used to motivate the proposed method, it remains unclear how the proposed algorithm addresses the claims beyond intuition.

**Requested Changes:**

While the method seems promising, it still remains a bit unclear to me how Equation 1 tackles the issue of learning from sub-optimal demonstrations. Addressing the following points can make improvements in that direction and strengthen my recommendation for acceptance:

- Intuitively it is believable that Equation 1 encourages the learning of a distance monotonic map, but it doesn’t feel enforced. The more formal intuition in Section A.2 helps but a better connection to the loss function of Equation 1 would be appreciated.
- It can make sense to me that the logarithm added in Section 3.4 can improve stability by reducing the variance of the regularization term, but no justification is provided.
- Figure 1 presents the reward of the RL agent against the average dataset reward. It would be useful to have the same average dataset rewards for Low, Medium and High in the results of Section 4.

Minor (not critical):

- Around Definition 3.1, it should be clarified that $d_S$ is the geodesic distance while $d_Z$ is the L2 norm. If this is not the case, please let me know and clarify in the paper.
- At the beginning of Section 4 there is a duplicated reference to PLAS.

---

> ### Author Response · Authors · 2026-01-15
>
> **W1 & RC1:**
>
> We agree with the reviewer on this concern. The primary goal of the paper is to tackle the problem of policy estimation in low-quality data regimes. As such, we built on recent work on value function approximation via metric learning, which allows us to avoid regularizations. We have presented the concept of DM maps and a proof of their optimality in a policy greedy sense to justify the use of these metric embeddings in the context of offline RL. The optimization described in Equation (1) is purposely kept simple to allow an easier optimization process. This, however, makes it quite difficult to formally prove that its convergence induces a DM map. The numerical experiment provided in Figure 3 (and the additional Figure 16 in the Appendix) does provide some empirical evidence of this.
>
> **W2:**
>
> Goal-conditioned offline RL is, in fact, a well-studied area. The main point of our this paper is to address the problem of low-quality demonstrations. The paper proposed by the reviewer (Transitive RL) addresses the distribution shift problem via a quantile regression instead of the Bellman update. This is akin to IQL.
>
> **RC2:**
>
> We thank the reviewer for the suggestion. We made this clearer in the revised paper. The negative term in the Objective is in general unbounded thus compromising the optimization process. The log term downplays the contribution of distant pairs of embeddings, allowing for an easier optimization. This is especially true for long-horizon tasks.
>
> **RC3:**
>
> Each environment required a different way of generating the different datasets. As such a unifying representation of each test condition might be difficult. We have, however, included additional details on the frequency of success of the collected trajectories for each environment and dataset in the Appendix in the revised version of the document.
>
> **RC4:**
>
> We added a clarification in the paper. Notice, however, how the definition of distance monotonic embedding holds independently of the notion of distance in both spaces.
>
> **RC5:**
>
> We apologize for the typo, we have updated this in the new version.

---

### Review · Reviewer_Xf2i · 2025-12-15

**Summary Of Contributions:**

This paper introduce MetricRL, a new offline IL method for learning policies in deterministic goal-conditioned & time-reversible MDPs. The paper introduced a desired latent space representation learning (distance monotonicity) technique that embed the state space into a latent space with a distance monotonic map. In addition, the authors propose to use weighted maximum likelihood for offline learning from the data. Experiments on grid world and its variants are presented, showcasing computational benefits, particularly its capabilities of learning from imperfect demonstration data.

**Additional Comments:**

The assumptions and the scope of this paper is extremely restrictive. The MDP is assumed to be time-reversible, discrete, with sparse rewards. Grid world is indeed one of the few that satisfies such requirements, which made me very concerned of the applicability of the proposed framework.

**Audience:**

Yes

**Audience Explanation:**

Learning sub optimal data is important in offline RL/IL.

**Claims And Evidence:**

No

**Claims Explanation:**

1. The main contribution of this paper is on learning from sub-optimal data. Perhaps I missed something, but I do not see the clear connection on how the latent representation learning benefit this aim. In fig.5 there's something interesting there but I don't see the connection clearly. Additionally, the experiments are largely grid worlds, which are too simple to be applicable in real world problems.

2. For the results, for Maze Large, the proposed method is not competitive for expert data. Why?

**Requested Changes:**

1. Can the authors explain clearly what's the contribution of the proposed representation learning to the learning of suboptimal data?

2. Can the authors provide detail explanation on using the weighted MLE (behavior cloning) loss for policy learning?

3. Can the authors explain why the chosen policy update loss can avoid OOD? I do not think behavior cloning alone avoid OOD completely, hence the claim made in the paper is inappropriate.

---

> ### Author Response · Authors · 2026-01-15
>
> **Q1:**
>
> The objective described in Equation (1) allows to learn a latent metric space where distances are correlated to the minimum number of actions needed to move from one state to another. In fact, distances are maximized between embeddings but locally chained together by forcing adjacent embeddings to have a constant distance. The optimization is computed solely on the dataset of demonstrations and does not make use of the max operator like in Bellman. This allows us to approximate the value function while avoiding the problem of distribution shift [1] often encountered in offline RL. Moreover, using a weighted maximum likelihood loss for the actor, restricts the learning process to in-distribution actions only. These two components are essential when dealing with sub-optimal data. A simple behavioral cloning loss, in fact, would result in the same policy that collected the data, thus sub-optimal. Moreover, both the Maze2D and Reach environments have a continuous state space and a continuous action space. The presented method is not restricted to discrete spaces.
>
> [1]: Levine, Sergey, et al. "Offline reinforcement learning: Tutorial, review, and perspectives on open problems." arXiv preprint arXiv:2005.01643 (2020).
>
> **Q2:**
>
> It is true that performances on the Maze2D Medium and Large environments are lower than a few baselines. For these experiments, we rely on the Minari dataset [1] which has an average success rate of more than 99% for the recorded trajectories. A simple Behavioral Cloning baseline is able to solve the task in most cases. Baselines that use BC regularization might have an unfair advantage. The main point of this paper is to avoid such regularizations.
>
> [1]: Omar G. Younis, Rodrigo Perez-Vicente, John U. Balis, Will Dudley, Alex Davey, and Jordan K Terry. Minari, September 2024. URL https://doi.org/10.5281/zenodo.13767625.
>
> **RC1:**
>
> We have rewritten the introduction to place more emphasis on the scope of the paper. The aim is to learn a policy under sub-optimality conditions. We propose to achieve this by avoiding the Bellman operator for estimating the critic and using an MLE policy to estimate the actor. We, additionally, present a property over this embedding that is needed to estimate an optimal policy (distance monotonicity) and propose the use of a symmetric space to induce a useful bias when the environment allows it (see Section 4.2). We would like to point out that the use of metric learning in the context of RL is not per se novel (see [1], [2], [3]). Here, we propose a simplification of the objective that is needed when optimization can be difficult (sub-optimal data) and study a property this map needs to satisfy in order to recover an optimal policy (distance monotonicity).
>
> [1]: Steccanella, L., & Jonsson, A. (2022). State representation learning for goal-conditioned reinforcement learning, ECML.
>
> [2]: Park, S., Rybkin, O., & Levine, S. (2023). Metra: Scalable unsupervised rl with metric-aware abstraction. arXiv preprint arXiv:2310.08887.
>
> [3]: Park, S., Kreiman, T., & Levine, S. (2024). Foundation policies with hilbert representations. arXiv preprint arXiv:2402.15567.
>
> **RC2:**
>
> As pointed out in Section 3.3, we pair the learned representation with a learned actor. The greedy policy, in fact, might be unfeasible to estimate when the action space is continuous, e.g., the Maze2D environment. While many implementations for this are possible, we rely on a modified version of a weighted maximum likelihood. This allows us to consider only in-distribution actions when training the policy, thus ensuring that the mode of the policy is an OOD action. Note, however, how the reweighting of the likelihoods plays a fundamental role in the training of the actor. Throughout the paper, in fact, we assume the distribution of the demonstrations to be non-optimal. A simple behavioral cloning policy would result in an actor with comparably sub-optimal behavior.
>
> **RC3:**
>
> We understand the reviewer’s doubt. We would like to clarify the meaning of OOD in this context. In the paper, we address the well-studied problem of out-of-distribution during training (or distribution shift) that dynamical programming methods suffer from in the offline RL framework. Essentially, when computing a value function via the Bellman operator, the max term leads to an overestimation of the value function that generally prevents effective learning, see [1] for an extended discussion on the topic. Furthermore, we avoid the use of policy constrain regularizations on the actor by restricting the learning on actions in the daset only (using a weighted maximum likelihood optimization). Constraining the policy from generating trajectories out of the dataset distribution (or out-of-distribution during testing) is outside of the scope of the paper.
>
> [1]: Levine, Sergey, et al. "Offline reinforcement learning: Tutorial, review, and perspectives on open problems." arXiv preprint arXiv:2005.01643 (2020).

---

### Review · Reviewer_1hWM · 2026-01-09

**Summary Of Contributions:**

The paper proposes a method to learn representations for goal-conditioned reinforcement learning. The method trains an embedding to put two spaces close together in Euclidean space if they are connected by an action, and further apart if they are not. The method is evaluated on standard low-dimensional goal conditioned control tasks and shows strong performance with sub-optimal data.
Crucially, the method employs a symmetry assumption, which forces the learned representation to approximate a metric space.

**Audience:**

Yes

**Audience Explanation:**

The paper is overall well written and easy to follow. The authors cleanly identify the lack of strong performance from suboptimal data as an important frontier in goal-conditioned training. While the limitations of the proposed method do not seem to be addressed as cleanly as I would prefer, I still believe that with better framing, the insights here can provide value for interested people in this field. Whether certain structure can and should be exploited is an interesting question, and while I believe the paper's current framing lacks some important nuance and clarification, it is still an interesting contribution.

**Claims And Evidence:**

No

**Claims Explanation:**

The core issue with the paper lies in the importance of the symmetry assumption. In the general RL setting, this assumption is simply not true. This greatly impacts the claims made in the introduction and positioning of the paper, as the proposed methods cannot be seen as general. Importantly, the method does not, as claimed, exploit symmetries, but rather presupposes them without any mechanism to verify that these symmetries actually exist.

This is also curious in the context of quasimetric learning. Symmetrizing quasimetric learning can only be successful if the underlying environment is indeed symmetric. However, it seems that the authors carefully picked environments in which this is (mostly) true. It would be very important to see this comparison in environments that violate the ground assumption, especially higher dimensional ones in which learning an explicit distance function might become intractable. The maze environments already provide versions with ant and humanoid robots, which would be a good start.

Finally, while the proposed method does outperform prior methods on low-quality data, it seems to suffer from poorer performance with high-quality datasets. This is a drawback that seems problematic in many application domains, e.g. robotics, where existing high-quality datasets strongly drive performance, and better (yet still not great) performance with suboptimal data would not necessarily justify a worse method on existing data. I am curious if this drop in performance is related to places in which the base assumption is indeed violated, or if there are other issues at play here.

To summarize, I believe that the symmetry assumption is too strong for a general method, and so either the paper needs to be expanded to address possible violations of this assumption, or the assumption (and resulting limitations) needs to feature prominently in the framing of the paper: introduction and research questions would need to be rewritten to highlight it.

**Requested Changes:**

See above. I believe there are two major directions in which the paper could be improved: either expanding the evaluation, or revisiting the framing.

---

> ### Author Response · Authors · 2026-01-15
>
> We agree with the reviewer on the limitation of the method's generality. We rewrote the introduction accordingly. The paper does not propose a general RL algorithm, but it is rather focused on the specific problem of learning a policy from a severely sub-optimal dataset of demonstrations. The use of a normed space to model the critic is inspired by previous works on the topic that have already pointed out this limitation (see [1], [2], [3]). In this regard, we propose a property on the embeddings of these spaces that might describe why they have been empirically useful in the context of goal-conditioned offline reinforcement learning, i.e., distance monotonicity. In the paper, we show that, while being a limiting assumption, modelling the critic via a normed space results in good performances for many benchmarks. While extensions to quasimetric spaces are possible (see [4], [5]), we believe this paper offers an interesting direction to a generally overlooked problem in offline RL, i.e., learning from low-quality data. As pointed out in Section 4.2, enforcing the symmetric bias can be beneficial in some cases when the main limiting factor comes from the data quality. As such, modelling the learned embedding via a Euclidean normed space allows us to effectively account for these symmetries automatically. We do, however, recognize that this might have been misleading.
>
> It is true that performances on the Maze2D Medium and Large environments are lower than a few baselines. For these experiments, we rely on the Minari dataset [6] which has an average success rate of more than 99% for the recorded trajectories. A simple Behavioral Cloning baseline is able to solve the task in most cases. Baselines that use BC regularization might have an unfair advantage. The main point of this paper is to avoid such regularizations. Note, additionally, that in the ablation of Section 4.2 we show how performances actually improve when synthetically symmetrizing the dataset for the Maze2D Large experiments for the Quasimetric RL baselines. Lastly, we believe one of the reasons why the robotics community has focused on high-quality datasets is because of a lack of methods focusing on the problem of extracting good behaviors from bad data, which is exactly what motivated this paper in the first place.
>
> [1]: Steccanella, L., & Jonsson, A. (2022). State representation learning for goal-conditioned reinforcement learning, ECML.
>
> [2]: Park, S., Rybkin, O., & Levine, S. (2023). Metra: Scalable unsupervised rl with metric-aware abstraction. arXiv preprint arXiv:2310.08887.
>
> [3]: Park, S., Kreiman, T., & Levine, S. (2024). Foundation policies with hilbert representations. arXiv preprint arXiv:2402.15567.
>
> [4]: Wang, T., & Isola, P. (2022). On the learning and learnability of quasimetrics. arXiv preprint arXiv:2206.15478.
>
> [5]: Bo Liu, Yihao Feng, Qiang Liu, and Peter Stone. Metric residual networks for sample efficient goal-conditioned reinforcement learning. arXiv preprint arXiv:2208.08133, 2022.
>
> [6]: Omar G. Younis, Rodrigo Perez-Vicente, John U. Balis, Will Dudley, Alex Davey, and Jordan K Terry. Minari, September 2024. URL https://doi.org/10.5281/zenodo.13767625.

---

### Author Response · Authors · 2026-01-15
**Changes in blue**

We thank all the reviewers for the valuable feedback. We have re-uploaded the paper with changes in blue.

---

### Decision · Action_Editor_Bi3k · 2026-03-02

**Recommendation:** Reject

**Audience:**

No

**Audience Explanation:**

Unfortunately, all reviewers agree that the scope of the paper is too narrow due to the restrictive assumptions.

Since the problem is an interesting one, I encourage the authors to explore some of the extensions mentioned by the reviewers:

 * The symmetry assumptions, being a strong one, should play a more prominent role in the framing of the paper
* The consequences of violating the assumptions should be explored further
* More complex environments should be considered in the experiments, even if they violate some of the assumptions
* Can the method made more robust, so that, when the data are actually high-quality, performance is not reduced significantly?

**Claims And Evidence:**

Yes

**Claims Explanation:**

The paper was rejected a first time mostly due to theoretical problems, although reviewers had already raised some concerns about the experiments. The theory was revised for this new submission, but some key statements were weakened.

This second round of reviews focused more on experimental results, which sufficiently support the claims.

**Resubmission Of Major Revision:**

The authors may consider submitting a major revision at a later time.